# Microcirculation in Hypertension: A Therapeutic Target to Prevent Cardiovascular Disease?

**DOI:** 10.3390/jcm12154892

**Published:** 2023-07-25

**Authors:** Damiano Rizzoni, Claudia Agabiti-Rosei, Gianluca E. M. Boari, Maria Lorenza Muiesan, Carolina De Ciuceis

**Affiliations:** 1Department of Clinical and Experimental Sciences, University of Brescia, 25121 Brescia, Italy; claudia.agabitirosei@unibs.it (C.A.-R.); marialorenza.muiesan@unibs.it (M.L.M.); carolina.deciuceis@unibs.it (C.D.C.); 2Second Division of Medicine, Spedali Civili di Brescia, 25123 Brescia, Italy; 3Division of Medicine, Spedali Civili di Brescia, Montichiari, 25123 Brescia, Italy; gianluca@boari.net

**Keywords:** microcirculation, peripheral circulation, remodeling, small resistance arteries, vascular biology

## Abstract

Arterial hypertension is a common condition worldwide and an important risk factor for cardio- and cerebrovascular events, renal diseases, as well as microvascular eye diseases. Established hypertension leads to the chronic vasoconstriction of small arteries as well as to a decreased lumen diameter and the thickening of the arterial media or wall with a consequent increased media-to-lumen ratio (MLR) or wall-to-lumen ratio (WLR). This process, defined as vascular remodeling, was firstly demonstrated in small resistance arteries isolated from subcutaneous biopsies and measured by micromyography, and this is still considered the gold-standard method for the assessment of structural alterations in small resistance arteries; however, microvascular remodeling seems to represent a generalized phenomenon. An increased MLR may impair the organ flow reserve, playing a crucial role in the maintenance and, probably, also in the progressive worsening of hypertensive disease, as well as in the development of hypertension-mediated organ damage and related cardiovascular events, thus possessing a relevant prognostic relevance. New non-invasive techniques, such as scanning laser Doppler flowmetry or adaptive optics, are presently under development, focusing mainly on the evaluation of WLR in retinal arterioles; recently, also retinal microvascular WLR was demonstrated to have a prognostic impact in terms of cardio- and cerebrovascular events. A rarefaction of the capillary network has also been reported in hypertension, which may contribute to flow reduction in and impairment of oxygen delivery to different tissues. These microvascular alterations seem to represent an early step in hypertension-mediated organ damage since they might contribute to microvascular angina, stroke, and renal dysfunction. In addition, they can be markers useful in monitoring the beneficial effects of antihypertensive treatment. Additionally, conductance arteries may be affected by a remodeling process in hypertension, and an interrelationship is present in the structural changes in small and large conductance arteries. The review addresses the possible relations between structural microvascular alterations and hypertension-mediated organ damage, and their potential improvement with antihypertensive treatment.

## 1. Introduction

Cardiovascular diseases are the main cause of death worldwide, resulting in around 18 million deaths per year (WHO, 2020) [1], and hypertension is considered as the main contributor to such a global burden [2]. Effective blood pressure (BP) control is a mandatory factor in order to improve hypertensive target organ damage and subsequent cardiovascular morbidity and mortality [2].

Functional, mechanical, and structural alterations of the microvasculature may be observed in patients with essential hypertension, even at very early stages [3,4,5], and contribute to the development of hypertension complications and cardiovascular prognosis [3,5]. In particular, in this setting, microvascular remodeling is a key event in triggering cardiovascular diseases [6,7].

The molecular mechanisms underlying the development of vascular remodeling are only partly understood. However, among the factors that may contribute to microvascular changes, an important role may be played by vascular inflammation with infiltration of inflammatory circulating cells and the release of inflammatory cytokines and chemokines, proliferative growth factors, as well as oxidative stress related to both a reduction in nitric oxide bioavailability and increased reaction oxygen species production [5,8,9]. These changes are promoted by mechanical, hemodynamic, or metabolic vascular insult and result in an altered vascular smooth muscle cell phenotype and the accumulation of the extracellular matrix [9]. The present review briefly summarizes the knowledge at present about microvascular remodeling and its role in hypertension-mediated organ damage and consequent cardiovascular events.

## 2. Microvascular Remodeling in Hypertension

Cardiovascular and metabolic diseases (in particular arterial hypertension) are very commonly associated with alterations in microcirculation [4,6,7]; morphological changes may involve small resistance arteries, arterioles, capillaries, and post-capillary venules [3,6,10]. BP is mainly influenced by vessel resistance and microcirculation is the key element of peripheral resistance regulation. As previously mentioned, microcirculation may be subdivided in small arteries, arterioles, and capillaries. Small resistance arteries are defined as arteries with a lumen diameter roughly between 350 and 100 µm. Their structure consists of an outer connective tissue adventitia, smooth muscle cell tunica media, and the endothelial layer [5,6]. Arterioles are vessels with internal diameters smaller than 100 µm, characterized by a single layer of smooth muscle cells. Small arteries and arterioles account for 45–50% of peripheral resistance and are defined as resistance arteries. These vessels have the capacity of contracting when transmural pressure increases, a feature called myogenic tone. A total of 23–30% of peripheral resistance is to be ascribed to capillaries (internal diameter less than 7 µm), whose walls are constituted only by a monolayer of endothelial cells [5,6,11,12,13].

It was proposed several years ago that increases in systemic vascular resistance may be a consequence of vascular smooth muscle increase and the concomitant narrowing of the arteriolar lumen [3,4,6,7,14]. Microcirculation remodeling occurs in primary hypertension [6,10]. Remodeling may be classified according to lumen and wall cross-sectional area changes [12,13]. Hence, remodeling might be inward when the internal diameter is reduced and outward when the lumen is unchanged [15,16]. On the other hand, hypertrophic remodeling occurs when the vessel wall material, i.e., media/wall cross-sectional area, increases along with media/wall thickness, whereas if the media/wall cross-sectional area does not change or is reduced, then remodeling is defined as eutrophic or hypotrophic, respectively [15,16]. Inward eutrophic remodeling is mainly observed in primary hypertension with a media thickness increase and lumen diameter decrease resulting in an unchanged media-cross sectional area [14,15,16,17]. On the contrary, hypertrophic remodeling, which may be outward or inward, has been shown in the secondary form of hypertension [17] and in cardiovascular diseases, such as diabetes mellitus [18,19], obesity [20,21], and metabolic syndrome [22], as well as in other endocrine diseases [23,24], independently, in the presence of an increase in BP increase. As result of the remodeling process, the wall-to-lumen ratio (WLR) and media-to-lumen ratio (MLR), are defined as the ratio between wall and media thickness, respectively, and lumen diameter increase.

Hypertensive patients [4,14], but also obese [25] or diabetic patients [4], present reduced basal and total capillary densities compared to the normotensive controls showing a structural anatomical rarefaction of capillaries rather than the presence of non-perfused vessels [4]. This reduction in total capillary density may consequently lead to an increase in peripheral resistances, thus negatively affecting tissue perfusion and nutrient delivery [26].

The time course of the development of hypertension in respect to the onset of microvascular alterations is not clear. An increase in the MLR of mesenteric small resistance arteries of spontaneously hypertensive rats may be present in a pre-hypertensive phase [4]. The data for humans are obviously difficult to obtain, since very few longitudinal data are presently available [4].

As previously mentioned, the hallmark of a microvascular remodeling process is an increased WLR and/or MLR. These parameters were initially measured in small resistance arteries by using wire or pressure micromyography, an in vitro ex vivo technique where a small artery is isolated from subcutaneous fat tissue and mounted on wire or glass cannulas and studied in isometric or isobaric conditions [27,28,29]. At present, non-invasive techniques are available [30], such as scanning laser Doppler flowmetry and adaptive optics, which allow the accurate visualization and measurement of retinal vessel morphological parameters at a high resolution providing similar information compared to the micromyographic system [31]. In particular, scanning laser Doppler flowmetry allows us to measure the external diameter of retinal arterioles in reflection images and to evaluate the internal diameter in perfusion images, according to a laser Doppler technique. A software-guided automatic comparison of the two images provides an estimation of the WLR [32,33]. With adaptive optics, the internal and external diameters of retinal arterioles are derived though an algorithm detecting the gradient of light’s reflection between the lumen of the vessel and the wall; these data are then used to calculate the WLR and wall cross-sectional area [30,34] (Figure 1). Compared to scanning laser Doppler flowmetry, adaptive optics detects retinal arteriole remodeling with a lower intra-observer variability [31] and provides very high-quality images [34] (Figure 1).

The cost, advantages, disadvantages, and indications/perspectives of the different available techniques for a non-invasive investigation of microvascular structure are summarized in Table 1 [30].

### 2.1. Pathophysiology of Microvascular Remodeling

Microvascular remodeling is the complex result of multiple factors [35]. Although the “primum movens” for the microvascular remodeling in primary hypertension is still unclear, underlying mechanisms include hemodynamic and mechanical changes, such as increased wall stress and prolonged vasoconstriction, but also hormonal factors as well as immune cell activation, inflammatory, and oxidative signaling [9]. The interaction among these factors induces alterations in the extracellular matrix (ECM), which involve integrins [36], tissue transglutaminase [37], as well as the activation of tissue metalloproteinases 2 and 9 activity with an increase in collagen deposition in the vascular wall [38]. Moreover, oxidative stress, which is, to date, defined as an imbalance between oxidants and antioxidants in favor of oxidants [39], is able to modulate ECM composition, cell growth and differentiation, and activate growth factors and pro-inflammatory genes [40].

In particular, a combination of inward vascular wall growth and apoptosis in the peripheral area of small arteries has been suggested in the development of inward eutrophic remodeling [41]. This process seems to represent a physiological protective mechanism against an increase in wall tension caused by increased BP, as wall tension is directly proportional to the lumen radius and inversely proportional to wall thickness according to Laplace’s law.

On the other hand, an altered myogenic tone, which is the artery’s ability to contract in response to an increased BP in order to protect distal vessels, has been hypothesized to induce hypertrophic remodeling and contribute to target organ damage [19]. In this context, reactive oxygen species (ROS), together with growth factors and mechanical injury, may switch the vascular smooth muscle cell phenotype from contractile to synthetic, thus inducing hypertrophy [42]. ROS are oxygen-derived molecules, such as superoxide (O^2−^), nitric oxide, peroxynitrite (OONO^−^), and hydrogen peroxide (H_2_O_2_), which are extremely important in the pathophysiology of hypertension and hypertension-associated target organ damage [40]. Many pro-hypertensive factors, including angiotensin II, aldosterone, endothelin 1, growth factors, immune factors, and salt, induce ROS production in endothelial cells, vascular smooth muscle cells, adventitia, and perivascular adipose tissue (PVAT), leading to vascular injury [40].

Indeed, the renin–angiotensin–aldosterone system (RAAS) and endothelin system, as well as catecholamines with sympathetic nervous system activation may play a crucial role in microvascular remodeling. Angiotensin II and endothelin exert vasoconstriction effects with diminished vasodilatation and may contribute to microvascular remodeling by regulating oxidative stress and the inflammatory mechanisms of vascular damage [7,43]. Angiotensin II elicits its deleterious effect through the angiotensin type-I receptor (AT1R) [44], a receptor that mediates pathways involved in trophic effects, hypertension, and inflammation, whereas the angiotensin 1–7/Mas receptor pathway seems to be protective [45].

Recently, PVAT dysfunction was demonstrated to be involved in microvascular alterations. PVAT seems to exert a key role in vasculature homeostasis; its activity results from the balance between the production of vasodilation and vasoconstriction substances [46]. In healthy conditions, PVAT regulates the contractile function of vessels releasing the PVAT-derived relaxing factor (ADRF), a molecule whose nature is still unknown. Adiponectin acts as a vasodilator through the activation of eNOS; it also reduces macrophage activation, vascular smooth muscle cell proliferation, oxidative stress, and increase insulin activity [46]. As the PVAT anti-contractile effect disappears when adiponectin receptors are blocked, adiponectin is considered a realistic candidate as ADRF. In particular, PVAT dysfunction occurs in systemic inflammatory states, such as obesity, where adiponectin activity and release are reduced [47]. On the other hand, under pathological conditions, such as obesity, diabetes, and hypertension, PVAT is also able to produce pro-inflammatory cytokines (IL-6, TNF-alpha, IL-8, MCP-1, TGF-Beta), prothrombotic factors (PAI-1), and vasoconstriction molecules, such as endothelin-1 and angiotensin II [48].

### 2.2. Role of the the Immune System in Microvascular Remodeling: Interaction with Hormonal Signals, the Sympathetic Nervous System, and PVAT

Hypertension is associated with a chronic inflammatory response due to immune cell accumulation in different hypertensive target organs, such as the kidneys, heart, brain, and blood vessels (including in the PVAT), thus exacerbating hypertension. This observation raised the hypothesis that hypertension might start as an overactivation of the central nervous system by the RAAS leading to a rise in BP with peripheral mechanical and oxidative damage [49,50]. Peripheral damage causing danger associated molecular patterns (DAMPs production) might be a trigger for immune responses as, normally, DAMPs activate the immune system and cause inflammation [50].

Animal models revealed the strong role of innate and adaptive immune cells in angiotensin II-induced hypertension [8]. In particular, adaptive immunity and, in particular, T-lymphocytes seem to play a major role [8]. Rats infused with angiotensin II elicited an increased production of T-helper (Th)-1 cytokines and a decrease in anti-inflammatory Th-2 cytokines, which may be restored by angiotensin II-receptor-1 blockers, but not by the vasodilator hydralazine, despite the similar effect on BP [51].

In order to clarify the role of the immune system in angiotensin II-mediated hypertension, many studies were conducted in immunodeficient mice. Ang II-dependent hypertension as well as vascular remodeling were demonstrated to be blunted in rats deficient in T- and B-lymphocytes [52]. T-regulatory cells (Treg), a particular subtype of T cells with the ability to suppress T-lymphocyte proliferation, seemed to also be involved in pressure regulation [53]. Indeed, rats with genetic hypertension (Dahl salt-sensitive strain) had reduced levels of Treg cytokines, less Treg cells, and more inflammatory cells in the aorta, as well as higher pressure compared with genetic normotensive rats [54]. Moreover, Treg-adoptive transfer induced a BP-lowering effect in angiotensin II rats [53]. In addition, Madhur et al. demonstrated that hypertension was associated with an increase in circulating Th17 cells and that IL-17 played an important role in Ang II-dependent hypertension [55]. Recently, it was reported that salt intake may stimulate RAAS and endothelin with consequent Th17 lymphocyte activation and a change in gut microbiome leading to intestinal wall inflammation, hence promoting microvascular damage and hypertension [56].

It has also been suggested that a splenic factor, the placental growth factor (PlGF), mediates a sympathetic stimulation of the spleen leading to costimulation and deployment in target organs of T cells promoting the onset of hypertension [57]. The spleen is innervated by sympathetic nervous system fibers that modulate immune cell responses through their neurotransmitters, thus generating a neuro-immune communication [58]. Particularly, there is a strong link between the central nervous system, RAAS, and immune system [49]. RAAS central stimulation causes an activation of peripheral T-lymphocytes and consequently vascular inflammation, further exacerbating BP [59]. This effect on T cells is centrally mediated, and it is associated with BP increase mediated by noradrenaline [60]. Noradrenaline further activates T-cell proliferation as the lymph nodes and spleen are innervated by sympathetic fibers [61].

Inflammation and oxidative stress mediated by RAAS cause PVAT dysfunction [46,47,62] and the impairment of endothelial function, thus contributing to pathophysiological hypertensive structural alterations Endothelial function is essential for microcirculation, as it regulates vascular tone, permeability, inflammation, and angiogenesis. Resistance artery endothelial function and nitric oxide (NO) availability might represent important factors involved in resistance artery remodeling, independently from cardiovascular risk factor exposure. This issue is specifically addressed elsewhere [63,64,65].

### 2.3. Prognostic Role of Microvascular Structural Alterations

The MLR of subcutaneous small resistance arteries has been suggested to represent the most prevalent and earliest form of arterial damage in essential hypertension [66] and it may be present very early on, even in the prehypertensive phase, at least in the experimental model [4].

From a physiological point of view, the reductions in the small artery and arteriole lumen are associated with an increase in flow resistance, even in condition of maximal dilatation, hence impairing the organ flow reserve [67,68,69]. Indeed, a relationship between the vasodilating capacity of coronary microcirculation and media-to-lumen ratio of subcutaneous small arteries was previously demonstrated [70] in patients with mild to moderate hypertension, thus suggesting that structural alterations in the subcutaneous vascular district (evaluated by wire micromyography in isolated vessels from fat biopsies obtained from the gluteal region) may be representative of similar alterations in coronary microcirculation, leading to a reduced coronary flow reserve [70,71].

Increased MLR of subcutaneous small resistance arteries has been positively related to hypertensive target organ damage, such as left ventricular hypertrophy [72,73] or carotid artery structure [73].

Importantly, the changes in small artery structure have a prognostic significance in both primary and secondary hypertension and in type-2 diabetes; indeed, an increased MLR is associated with a reduced event-free survival for cardiovascular events in high-risk patients [3,74] as well as in medium-risk ones [3,75,76]. Moreover, the presence of hypertrophic remodeling seems to be associated with an even worse prognosis compared to eutrophic remodeling [77,78]. More details about the prognostic role of the structural alterations in subcutaneous small resistance arteries evaluated by micromyography are reported in reference [3].

On the other hand, WLR evaluated by scanning laser Doppler flowmetry was proved to increase in patients with hypertension and cerebrovascular disease [32], and to directly be associated with urinary albumin excretion [79]. Likewise, an increased WLR evaluated with adaptive optics correlated with age [80,81], BP [34], and may be improved by a reduction in BP values by hypertensive treatment [46]. Most recently, the prognostic role of WLR evaluated with adaptive optics was demonstrated [80]. The event-free survival was significantly worse in 230 normotensive subjects and hypertensive patients with a baseline WLR higher than the median value of the population (0.28) according to Kaplan–Meier survival curves and multivariate analysis (Cox’s proportional hazard model) [82]. The evidence was confirmed after restricting the analysis to cardiovascular events, excluding deaths and neoplastic diseases [82]. Therefore, the structural alterations of retinal arterioles evaluated by adaptive optics may predict total and cardiovascular events [82].

Most recently, it was also demonstrated that patients with coronary microvascular dysfunction, defined as the presence of a reduced myocardial flow reserve (≤2), evaluated by dynamic single-photon-emission computed tomography (SPECT), had higher rates of adverse outcomes that those without it [83]. However, it is not presently known whether capillary rarefaction may be related to cardiovascular events [3,4].

### 2.4. Possible Prevention/Regression of Microvascular Remodeling

The subsequent question to be answered is whether we should aim at correcting the structure of resistance vessels in the treatment of hypertension and whether this can affect the prognosis. Indeed, several drugs have been demonstrated to improve microvascular structure and therefore reduce the MLR, such as drugs inhibiting RAAS and dihydropyridinic calcium channel blockers, whereas diuretics and beta-blockers did not seem to have any relevant effects [5,7,84].

There is clear evidence that angiotensin-converting enzyme (ACE)-inhibitors and angiotensin-receptor blockers (ARBs) appear to be more effective than atenolol in terms of microvascular protection also in patients with diabetes mellitus [85,86,87]. Along this line, the direct renin inhibitor aliskiren improved microvascular structural alterations to a similar extent, compared with ramipril, in mild hypertensive patients with type-2 diabetes mellitus [88].

An improvement of microvascular remodeling was also demonstrated in the retinal vasculature. In patients with hypertension, aliskiren plus valsartan ameliorated ameliorate the WLR of retinal arterioles measured non-invasively by scanning laser Doppler flowmetry [89]. The combination of lercanidipine and enalapril was more effective in reducing the retinal arteriole WLR compared to the combination of lercanidipine and hydrochlorothiazide [90]. Using adaptive optics, the normalization of the retinal arteriole structure was observed after chronic and effective antihypertensive treatments [80]. On the contrary, a short-term reduction in BP only led to an increase in the internal diameter of retina arterioles with no changes in the wall thickness or wall cross-sectional area [80].

A complete regression of the remodeling process is difficult to obtain for patients with hypertension-mediated organ damage (i.e., left ventricular hypertrophy) or when comorbidities, such as diabetes mellitus, are present [3,4,7,84]. Usually, in such clinical situations, an improvement, but not a full normalization of the MLR of subcutaneous small resistance arteries, has been observed, despite effective BP reduction.

Several specific mechanisms may contribute to the beneficial effects of some drugs on small artery structure and outcomes compared with other pharmacological strategies. Buus NH et al. observed that, in essential hypertension, after one year of treatment with the ACE-inhibitor perindopril, the coronary flow reserve improved with the normalization of small subcutaneous arteries’ structure [91]. In this study, in the parallel group treated with the beta-blocker atenolol, coronary flow was reduced and the small subcutaneous artery structure remained unchanged, despite a similar reduction in BP [91]. Therefore, it can be speculated that the beneficial effect on the hypertensive target organ damage of these drugs may be also occur to their ability to improve microvascular structure. Buus NH et al. [92] also demonstrated that MLR represents an independent predictor of cardiovascular events, beyond the extent of BP reduction, in a cohort of moderate-risk essential hypertensive patients not only including untreated patients, but also hypertensive patients during long-term effective treatment. This suggests that the assessment of the microvascular structure may also be important in treated patients, since it can allow us to identify those who may benefit from a more aggressive treatment and further risk reduction.

In addition, since vascular damage in hypertension is also caused by the inflammation, oxidative stress, and immune system activation, drugs selectively modulating these pathways can represent a potential and interesting strategy of treatment in the near future.

In patients with Conn’s syndrome (primary aldosteronism due to an adenoma of the adrenals), the presence of marked or persistent vascular remodeling, as indicated by a higher MLR of subcutaneous small resistance arteries, was associated with reduced chances of BP normalization in the long-term follow-up after adrenalectomy. Thus, the severity of structural alterations in subcutaneous small resistance arteries can predict the clinical outcomes for these patients with secondary hypertension [93]. Therefore, structural alterations of small arteries can possibly be considered an important intermediate endpoint for the evaluation of the efficacy of antihypertensive treatment [4].

Antihypertensive treatment with ACE inhibitors seems to improve capillary rarefaction [90,94]; although, some methodological caveats were raised [95]. In any case, it has not yet been established whether a reversal of capillary rarefaction is associated with an improved prognosis [4].

## 3. Interrelationship between Microvascular and Macrovascular Remodeling

The large arteries are not only the target of high BP, but a determining factor in the pathogenesis of hypertension, particularly of isolated systolic hypertension and the increase in pulse pressure typical when aging [96,97]. The large elastic arteries allow the conduction of blood from the heart to the resistance arteries, but also for the transformation of the pulsatile flow, generated by cardiac activity, into the continuous flow observed in peripheral circulation. Part of the energy produced by the left ventricular systole is used for the stretching of arteries and it is stored in their walls. During diastole, this energy recoils the aorta and squeezes out blood into microcirculation, ensuring a continuous flow to tissues [98,99]. In essential hypertension, remodeling occurs not only in the small arteries, but also during macrocirculation [100,101]. Large conduit arteries (such as the aorta and its branches) develop arteriosclerosis of the media. This remodeling is characterized by an increase in intima-media thickness with a lumen enlargement of proximal elastic arteries, and it is a compensation mechanism used to normalize circumferential wall stress [99]. Elastic fibers in the media become thinner and frail, undergoing fragmentation with a parallel increase in collagen deposition and a consequent reduction in distensibility [96,100]. Aging and BP are the two main determinants of arterial stiffness, which is also influenced by diabetes mellitus, obesity, and metabolic syndrome. In the Framingham Heart Study, increased large-artery stiffness, measured as pulse wave velocity, and aortic root enlargement were observed to be associated with a higher risk of cardiovascular disease [5,102]. Pulse wave velocity correlates with hypertensive heart disease and microalbuminuria [103,104].

In fact, arterial stiffness has unfavorable consequences from a hemodynamic point of view. The increase in the speed of the incident and the reflected wave causes them to merge earlier, in the first part of the systole, increasing the systolic BP (with an increased afterload), reducing the diastolic BP (with a reduction in myocardial blood flow), and increasing the pulse pressure [100]. In addition, with the elastic arteries becoming more rigid than the muscular ones, a reversion in the normal center–periphery stiffness gradient occurs, which is mainly responsible for the reflection of the sphygmic wave [4,100]. Hence, the reflection site moves more distally, and the reflected wave is decreased, increasing the peripheral transmission of a large incident wave that exposes the peripheral arteries and arterioles to harmful levels of pressure plasticity, contributing to alterations of microcirculation and, in the end, of the nutrition and oxygenation of peripheral tissues (heart, brain, kidneys, and limbs), as well as the elimination of waste products [96,100].

It is still difficult to establish a temporal or linear relationship between small- and large-artery alterations in essential hypertension. The most likely relationship is actually a cross-talk between micro- and macrocirculation that may trigger a vicious cycle [96,100]. Hypertension causes the degeneration of large arteries and stiffness, with consequent higher central systolic and pulse pressures, and microvascular alterations, such as eutrophic remodeling, impaired vasodilatation, and microvascular rarefaction. Small-artery remodeling and rarefaction increase total peripheral resistance and amplify mean BP [96,100]. Indeed, in essential hypertension, the MLR of small resistance arteries and pulse wave velocity are independent determinants of central systolic BP [105]. Moreover, many indices of large-artery stiffness (e. g. pulse pressure, pulse wave velocity) are associated with indices of microvascular damage (WLR and MLR, respectively) [106,107].

Drugs that improve microvascular structure are particularly effective, also in reducing central BP, thus probably providing an additional benefit [4,62,69,74,75], most likely by slowing down the reflection of BP waves from distal reflection sites, close to microcirculation [4,96,100,103,108,109].

The pathophysiological consequences of the regression of small-artery remodeling might be: better BP control, taking advantage of reduced vascular reactivity [4,69]; an improvement of the organ flow reserve, in particular, in the heart [4,69,92]; and an effective reduction in central BP [4,96,100,103,108,109].

## 4. Conclusions

In hypertensive microcirculation, internal media or total wall thickness were increased in relation to the internal lumen, and this alteration contributed to the increase in peripheral resistance we observed [4]. Microvascular structural alterations might also impact the organ flow reserve [3,4,10,69,70], and this may play a role in the maintenance/progressive worsening of hypertensive disease. Therefore, an increased MLR in small resistance arteries may forecast the development of hypertension-mediated organ damage/cardiovascular events, as well as complications of the disease [77,78].

In order to allow for a wider application of the evaluation of microvascular morphology, we need non-invasive techniques to better stratify cardiovascular risk and to better evaluate the effects of antihypertensive therapy [4,30]. Techniques that allow for an evaluation of retinal artery morphology, such as scanning laser Doppler flowmetry or adaptive optics, seem to be a promising approach [30], as also stated in the 2023 ESH Guidelines for the management of arterial hypertension [110].

In conclusion, the evaluation of microvascular structure is progressively moving from bench to bedside [3,4,30,44], and can, in the near future, represent an evaluation to be performed in the majority, if not in all, hypertensive patients [3,4,30]. The most recent demonstration of the possible prognostic relevance of non-invasive measures of microvascular structure by adaptive optics represents a relevant contribution; although, this evidence has to be confirmed by other studies. In addition, we need a similar demonstration of the prognostic relevance of the changes in the indices of microvascular structure evaluated non-invasively and observed during antihypertensive treatment.

## Figures and Tables

**Figure 1 jcm-12-04892-f001:**
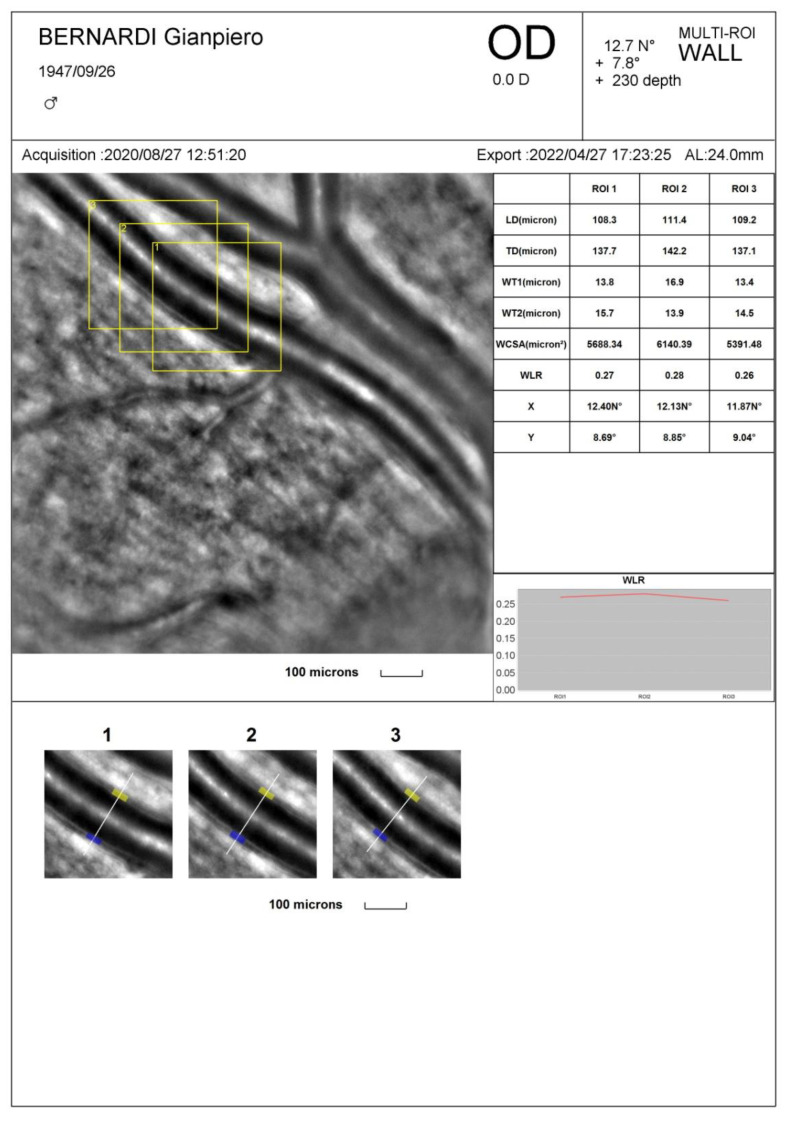
Evaluation of the morphology of retinal arteries by adaptive optics. Example of images obtained with adaptive optics (AO camera, Rtx-1, Imagine Eyes, Orsay, France) (**left** and **bottom**) and measurement of morphological parameters using a dedicated software (**right**). From reference [4].

**Table 1 jcm-12-04892-t001:** Cost, advantages, disadvantages, and indications/perspectives of the available techniques for a non-invasive investigation of microvascular structure. From reference [30].

Technique	Cost ($)	Advantages	Disadvantages	Indications/Perspectives
Forearm plethysmography	1500–2000	Relatively inexpensive	Locally invasive, needs experience	Research purposes, not for extensive clinical use
Intravital microscopy/Glycocalyx	3000–5000	Non-invasive	No prognostic data	Research purposes, not for extensive clinical use, though adoption in specific settings (i.e., critical care) is being proposed
Nailfold videocapillaroscopy	10,000–20,000	Non-invasive	No prognostic data	Research purposes in the cardiovascular setting, commonly used in rheumatology. Possible future demonstration of prognostic usefulness of such an approach might extend its clinical application to patients at elevated cardiovascular risk.
SLDF (Heidelberg Retina Flowmeter)	30,000–40,000	Some prognostic data available/possibility to assess endothelial function	No more in the market	Research purposes; potential for an extensive future clinical application in the cardiovascular field if technically developed by the producer
Adaptive optics cameras	130,000–160,000	Reliable/possibility to assess endothelial function	Cost, no prognostic data	Commonly used in ophthalmology for specific clinical purposes. Potential for an extensive future clinical application in the cardiovascular field
Dynamic Vessel Analyzer (AV ratio)	5000–6000	Some prognostic data, relatively easy to perform	Cost, some limitations in reliability	Commonly used in ophthalmology for specific purposes
OCTA	20,000–70,000	Useful vascular information on choroidal microvessels in hypertensive patients, plenty of data provided.	Imaging artefacts, high acquisition costs, role in the cardiovascular field still under evaluation	Commonly used in ophthalmology for specific purposes
Techniques for the evaluation of topological changes in the retinal vascular architecture or fractal dimensions	30,000–50,000	Some prognostic data (fractal dimensions)	Not standardized	Only research purposes
EndoPAT	2500–4000	Easy to perform, cost	Limited reliability	Only research purposes
Laser Doppler Flowmetry (skin)	1500–20,000	Easy to perform, cost	Limited reliability	Only research purposes

## Data Availability

Not applicable.

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
