# Peer review of "Microcirculation in Hypertension: A Therapeutic Target to Prevent Cardiovascular Disease?"

_jcm, 2023, doi:10.3390/jcm12154892_

Round 1

Reviewer 1 Report

Dear author.

Your manuscript is interesting and potentially important as it addresses the link between microcirculation and hypertension, which may help for new therapeutic targets. However, there are some issues that need to be addressed.

  1. Some statements in your manuscript are not supported by relevant references. For instance: “Small arteries and arterioles account for 45- 50% of peripheral resistance… 23-30 % of peripheral resistance is to be ascribed to capillaries”. Please provide appropriate citations for these statements, following the journal’s guidelines.
  2. Some sentences in your manuscript are too long and complex, making them difficult to follow for the reader. For instance: “In particular, Scanning Laser Doppler Flowmetry allows to measure the external diameter of retinal arterioles in reflection images and to evaluate the internal diameter in perfusion images according to a laser Doppler technique then providing an automatic estimation of WLR (29,30).” Please simplify and shorten these sentences, and use transitions and connectors to improve the coherence and flow of your text.
  3. Your manuscript is a bit biased as it provides only one aspect of the link between microcirculation and hypertension, namely the structural aspect of microcirculation, such as remodeling and wall changes. However, this topic is much broader and involves other aspects of physiology, such as endothelial function, signaling, immune interplay, and genetic factors, that are not covered in your manuscript. For instance, you should provide a link between hypertension and microcirculatory vessels from the perspective of endocrine or paracrine factors, as these factors modulate the endothelial function and signaling that are crucial for vessel remodeling. Endothelial function is essential for microcirculation, as it regulates vascular tone, permeability, inflammation, and angiogenesis, but your manuscript does not cover any of these aspects. You should also discuss how signaling pathways, such as nitric oxide, reactive oxygen species, angiotensin II, endothelin-1, and others, mediate the effects of mechanical forces and metabolic stimuli on microvascular remodeling. Moreover, you should address how immune cells and cytokines interact with endothelial cells and vascular smooth muscle cells to modulate vascular inflammation and remodeling in hypertension. Furthermore, you should consider how genetic factors influence the susceptibility and response of microcirculation to hypertension and its complications. Please provide a more comprehensive and in-depth review of the link between microcirculation and hypertension, including more aspects and perspectives.

In summary, your manuscript needs significant elaboration and refinement. Please address these major concerns and revise your manuscript accordingly.

Sincerely,

Reviewer

Some sentences in your manuscript are too long and complex, making them difficult to follow for the reader.

 However, there are some issues that need to be addressed before publishing

In summary, your manuscript needs significant elaboration and refinement before it can be considered for publication. 

Author Response

The Authors thank the Reviewer for his/her helpful suggestions.

1) Some statements in your manuscript are not supported by relevant references. For instance: “Small arteries and arterioles account for 45- 50% of peripheral resistance… 23-30 % of peripheral resistance is to be ascribed to capillaries”. Please provide appropriate citations for these statements, following the journal’s guidelines.

We have added some additional specific references.

 Bohlen HG. Localization of vascular resistance changes during hypertension. Hypertension. 1986;8:181–3.

Borders JL, Granger HJ. Power dissipation as a measure of peripheral resistance in vascular networks. Hypertension 1986; 8:184-91.

Christensen KL, Mulvany MJ. Location of resistance arteries. J Vasc Res 2001; 38:1–12.

2) Some sentences in your manuscript are too long and complex, making them difficult to follow for the reader. For instance: “In particular, Scanning Laser Doppler Flowmetry allows to measure the external diameter of retinal arterioles in reflection images and to evaluate the internal diameter in perfusion images according to a laser Doppler technique then providing an automatic estimation of WLR (29,30).” Please simplify and shorten these sentences, and use transitions and connectors to improve the coherence and flow of your text.

We have tried to modify some long sentences

3) Your manuscript is a bit biased as it provides only one aspect of the link between microcirculation and hypertension, namely the structural aspect of microcirculation, such as remodeling and wall changes. However, this topic is much broader and involves other aspects of physiology, such as endothelial function, signaling, immune interplay, and genetic factors, that are not covered in your manuscript. For instance, you should provide a link between hypertension and microcirculatory vessels from the perspective of endocrine or paracrine factors, as these factors modulate the endothelial function and signaling that are crucial for vessel remodeling. Endothelial function is essential for microcirculation, as it regulates vascular tone, permeability, inflammation, and angiogenesis, but your manuscript does not cover any of these aspects. You should also discuss how signaling pathways, such as nitric oxide, reactive oxygen species, angiotensin II, endothelin-1, and others, mediate the effects of mechanical forces and metabolic stimuli on microvascular remodeling. Moreover, you should address how immune cells and cytokines interact with endothelial cells and vascular smooth muscle cells to modulate vascular inflammation and remodeling in hypertension. Furthermore, you should consider how genetic factors influence the susceptibility and response of microcirculation to hypertension and its complications. Please provide a more comprehensive and in-depth review of the link between microcirculation and hypertension, including more aspects and perspectives.

Two additional parts was added (Pathophysiology of microvascular remodeling and Role of the immune system in microvascular remodeling: interaction with hormonal signals, sympathetic nervous system and PVAT, dealing with endothelial function, signalling, immune interplay, and genetic factors.

Changes made to the manuscript were highlighted in red,

Reviewer 2 Report

Manuscript is well drafted and appropriately covered  important areas;  existing evidence on the presence of microcirculatory remodeling in patients with hypertension, their potential role in the pathogenesis of hypertension mediated organ damage/dysfunction,, as a predictor of CV outcomes and potential of noninvasive methods of measuring remodeling and effect of antihypertensive agents on progression/regression of microcirculatory remodeling.  

I think topic of review will be of interest to wider audience in providing insights about greater understanding of pathogenesis of Hypertension related target organ damage and CV events. 

Author Response

The Authors thank the Reviewer for his/her helpful suggestions.

Manuscript is well drafted and appropriately covered  important areas;  existing evidence on the presence of microcirculatory remodeling in patients with hypertension, their potential role in the pathogenesis of hypertension mediated organ damage/dysfunction,, as a predictor of CV outcomes and potential of noninvasive methods of measuring remodeling and effect of antihypertensive agents on progression/regression of microcirculatory remodeling. 

I think topic of review will be of interest to wider audience in providing insights about greater understanding of pathogenesis of Hypertension related target organ damage and CV events.

The Authors thank the Reviewer for his/her favourable consideration of our efforts.

Reviewer 3 Report

The review is interesting and offers many ideas for future research perspectives.

The working group has consolidated scientific experience on the subject, so I think that they must to correct the Fig. 1 on page 4: the patient's name must be absolutely obscured, keeping the initials.

Correct the wrong year of publication (if any) that appears on each page of the review: J. Clin. Med. 2021 instead of 2023, probably not dependly from you.

Author Response

The Authors thank the Reviewer for his/her helpful suggestions.

The review is interesting and offers many ideas for future research perspectives.

The working group has consolidated scientific experience on the subject, so I think that they must to correct the Fig. 1 on page 4: the patient's name must be absolutely obscured, keeping the initials.

The Authors thank the Reviewer for his/her favourable consideration of our effort. The name of the patients is now obscured. It was a mistake made during image processing (from PowerPoint to Microsoft Word)

Correct the wrong year of publication (if any) that appears on each page of the review: J. Clin. Med. 2021 instead of 2023, probably not depending on you.

Unfortunately, we have no control on this issue, since such a date is generated by the submission system, probably during file conversion.